# Coupling and Coordinating Relationship between Agricultural Eco-Efficiency and Food Security System in China

**DOI:** 10.3390/ijerph20010431

**Published:** 2022-12-27

**Authors:** Ruixue Wang, Jiancheng Chen, Minhuan Li

**Affiliations:** School of Economics & Management, Beijing Forestry University, Beijing 100091, China

**Keywords:** agricultural eco-efficiency, food security, coupling coordination

## Abstract

Carbon peaking, carbon neutrality goals and food security are the basis of sustainable development, and exploring the coordination relationship between China’s agricultural eco-efficiency and food security system has a major significance for the implementation of relevant strategies. This paper is based on collaboration research on the synergistic relationship between agricultural eco-efficiency and food security systems using methods such as entropy weight method, coupling coordination model, spatial autocorrelation model, etc., revealing the evolution-driven mechanism of the coupling coordination degree. This study found that a higher level of coupling coordination always occurs in those areas with high standard farmland construction and large grain production scale, while economically developed areas appear to have a lower overall coordination level limited by endowment constraints and division of labor in development planning. It shows a positive spatial correlation in terms of geographical distance between agricultural eco-efficiency and food security, and the positive spillover effect gradually increases but is not strong overall. China should combine regional resource endowment and development planning, pay attention to the improvement of large-scale and standardized agricultural production, continue to strengthen the development of clean agricultural production, and achieve food security under the constraints of the carbon peaking and carbon neutrality goals.

## 1. Introduction

Balancing ecological environmental protection and sustainable economic and social development is a hot topic in the world. The Sixth Assessment Report of the Intergovernmental Panel on Climate Change (IPCC) conducted a scientific assessment of sustainable agricultural development and food security in the context of climate change, discovered that man-made greenhouse gas emissions are important causes of global warming and land use changes, which in turn affect sustainable agricultural development and food security [1,2]. The Food and Agriculture Organization of the United Nations (FAO) released a report at the 26th United Nations Climate Change Conference, stating that carbon emissions from the agrifood system accounted for 31% of global anthropogenic carbon emissions in 2019 [3]. It can be seen that climate change has a profound impact on agricultural development, and carbon emissions from the agro-food system are also one of the factors affecting climate change.

Agriculture is a national basic industry, which not only undertakes the function of natural ecological regulation, but also plays an important role in ensuring the food security of the country and the world [4]. As a climate-sensitive region and a populous country, China is facing increasing risks. In 2020, China’s total agricultural carbon emissions accounted for 20%, of which the carbon emissions caused by agricultural energy utilization, agricultural material input, and rice planting accounted for 14.21%, 26.38%, and 25.95%, respectively [5]. It can be seen that China’s agricultural carbon emissions have huge potential for emission reduction, and play an important role in the realization of China’s carbon peak and carbon neutral goals. Promoting green and low-carbon development in agriculture is an issue that China has repeatedly emphasized. While ensuring food security, China is committed to achieving green transformation in agriculture [6]. Therefore, the focus of China’s agricultural low-carbon sustainable development is to explore the path of food security on the basis of improving China’s agricultural eco-efficiency.

The purpose of this study is to investigate the relationship between agricultural eco-efficiency and the food security system in China. The main contributions of this paper are as follows. First, the international research is extremely limited on the relationship between agricultural eco-efficiency and food security [7,8,9]; this study fills the gap in this regard. At the same time, this study attempts to bring China’s agricultural eco-efficiency and food security into the same framework, discusses the level of coupled and coordinated development of them, clarifies the characteristics of the current situation, and analyzes the spatial correlation. This study has practical significance for China’s agricultural development and provides a theoretical reference for the green and sustainable development of agriculture in other developing countries.

The remainder of this paper is organized as follows. The second section introduces the research progress on agricultural eco-efficiency and its correlation. The third section introduces the data sources and research methods of this paper. The fourth section presents the results and discussion of the empirical research. The fifth section is the conclusion and policy recommendations from the results.

## 2. Literature Review

### 2.1. Agricultural Eco-Efficiency and Its Estimation

Ecological efficiency refers to the ratio of the added value of activities to the increased environmental impact, which was first proposed by German scholars Schaltegger and Sturm in 1990 [10]. Since then, the World Business Council for Sustainable Development (WBSCD), the Organization for Economic Co-operation and Development (OECD), and the United Nations Conference on Trade and Development (UNCTAD) have successively conducted in-depth research on eco-efficiency, and defined its concept from different perspectives [11,12]. Agricultural eco-efficiency is the expansion and application of the concept of eco-efficiency in the agricultural field. There is no clear definition yet. It usually refers to the ratio of the ideal minimum carbon emission to the actual carbon emission under the given input and output conditions. To a certain extent, it reflects the level of agricultural productivity under the constraints of established carbon emissions, and it is the embodiment of ecological concepts in the agricultural field [13,14].

At present, the methods for measuring agricultural eco-efficiency mainly include ratio method, life cycle assessment method, ecological footprint analysis method, data envelopment analysis method, etc. [15,16,17]. As an extension of the data envelopment analysis (DEA) method, the undesirable output slacks-based measure (SBM) model not only inherits the characteristics of the traditional DEA method to avoid the deviation and influence of efficiency evaluation caused by differences in radial and angle selections, but also takes into account the impact of undesired output factors, making the calculation of eco-efficiency results more objective and comprehensive [18].

On the basis of considering the characteristics and connotations of agricultural eco-efficiency, the undesirable output SBM model uses agricultural carbon emissions as an undesired output to construct an input and output index system, and uses a non-radial and non-angular model to measure and calculate the agricultural eco-efficiency, assessing the characteristics of agricultural eco-efficiency by region and year [19,20]. It provides effective countermeasures for agricultural carbon sequestration and emission reduction by measuring the dynamic changes, spatial correlations, and influencing factors of agricultural eco-efficiency [21].

### 2.2. Agricultural Eco-Efficiency and Food Safety Studies

Since the 1990s, economic eco-efficiency has gradually become a research focus of scholars with the concept of sustainable development. The research on agricultural eco-efficiency mainly focuses on the evolution of spatial patterns and the quantitative description of regional distribution, focusing on considering the negative external effects of pesticides and fertilizers from the perspective of environmental pollution [22,23,24]. At present, the academic circle mainly studies the agricultural eco-efficiency of China from two perspectives. The first is to measure the changing trend of China’s agricultural eco-efficiency and analyze the influencing factors. The approach is to calculate agricultural eco-efficiency by region, province, and year [25,26], to evaluate the potential of agricultural carbon emission reduction in each region, and explore the reasons for the loss of agricultural eco-efficiency in different regions by combining redundant indicators. This includes the spatial spillover effects and characterization of the spatial characteristics to explore the improvement path of regional agricultural eco-efficiency [27,28,29]. The results show that in the past few decades, China’s agricultural eco-efficiency has shown a fluctuating upward trend, with significant regional differences and serious labor redundancy. Eco-efficiency values showed significant spatial agglomeration characteristics, and the hotspots were mainly distributed in the eastern region.

The second is to explore the effect of ecological agriculture on food security, and to seek a healthy and long-term sustainable method for food production. For example, da Silva JT et al. calculated the ecological footprint of food production activities through an econometric model [30], judged the ecological sustainability based on the results, and emphasized the importance of increasing food imports to alleviate the pressure on agricultural resources and the environment [31]. Other studies focus on the impact of ecological security issues such as water resource constraints and biodiversity loss on food security, emphasizing farmland ecological governance, crop variety optimization, and improvement of cultivated land fertility to achieve the dual goals of agricultural eco-efficiency and food security [32,33]. At the same time, some studies have attributed the risk of China’s grain production to the imperfect food ecological security system, indicating that the key to ensuring food security lies in improving agricultural eco-efficiency [34,35].

However, relevant research focuses on exploring the ecological problems caused by food production, focusing on the ecological perspective and the current situation of agricultural ecological development. There are few studies on the relationship between agricultural eco-efficiency and food security, and the empirical results are different. It mainly focuses on the coupling and coordination between agricultural socio-economic factors and food security, and analyzes the impact of water, soil, energy, and other utilization rates on the sustainability of food security [36,37,38]. Some studies believe that in order to ensure food security, China needs to appropriately reduce its food self-sufficiency rate, and ease the pressure on agricultural resources and ecological environment by increasing imports and reducing production [39]. Other studies have shown that the green transformation of agricultural ecology under the goal of food security can be achieved through high standard farmland assumptions, mechanization, and technological development, and the win–win situation of improving agricultural eco-efficiency and food security can be promoted [40].

Although China is a large country in terms of carbon emissions and food demand, there are few studies on the coordination relationship between agricultural eco-efficiency and food security. With climate change in recent years leading to rising global temperatures and heavy burdens on resources, preventing the ecological risks of food security, enhancing the sustainable development of food production, and realizing the coordinated development of ecological agriculture and food security should be important topics in current research. In this regard, we first estimated the level of agricultural eco-efficiency and food security in each province in China, and then measured the relationship between them.

## 3. Methodology and Data

### 3.1. Methodology

Based on theoretical analysis, firstly, agricultural carbon emissions were calculated through the emission factor method, and then the agricultural eco-efficiency was calculated through the SBM-Undesirable model. Second, the comprehensive index evaluation method was used to measure the level of food security. Finally, the coupling coordination degree of agricultural eco-efficiency and food security was measured by the coupling coordination degree model, and the spatial correlation model was used to explore the spatial correlation. The specific steps are shown in Figure 1.

#### 3.1.1. Calculation of Agricultural Carbon Emissions

This paper draws on Li Z’s method [41] and uses the emission factor method to measure agricultural carbon emissions. The emission factor method (Emission-Factor Approach) is one of the carbon emission estimation methods proposed by IPCC, and it is also a method widely used in academic circles. For each carbon emission source of the research subject, its activity data and carbon emission factor are constructed, and the product of the two is taken as the carbon emission estimate of the research subject. Based on carbon sources, such as chemical fertilizers, pesticides, agricultural films, diesel fuel, irrigation, and tillage, this paper constructs a calculation method for agricultural carbon emissions:(1)$$E=\sum E_{i}=\sum T_{i}\times \delta_{i}$$

In the formula, *E* is the total amount of agricultural carbon emissions, $E_{i}$ is the *i* carbon emission of type *i* carbon source. $T_{i}$ means the *i*-th source. $\delta_{i}$ is the carbon emission coefficient of type *i* carbon source. According to the comprehensive consideration of relevant literature, the summary of agricultural carbon emission sources and coefficients is shown in Table 1.

#### 3.1.2. Estimating Agricultural Eco-Efficiency

The paper uses the undesirable output SBM model to measure the agricultural eco-efficiency. The model is evolved based on the traditional DEA method, taking into account the slackness of input and output and the efficiency level of “bad output”, effectively reducing the result deviation caused by the difference in radial and angular selection [45]. Combined with the actual situation of agriculture, this paper first constructs a production possibility set including input, expected output, and undesired output.

Suppose there are n decision-making units in the sample, each decision-making unit has m input variables x, S kinds of outputs, which include $S_{1}$ expected output $y^{g},$ and $S_{2}$ expected outputs $y^{b}$, the vector is expressed as $\in R^{m}$, the expected output $y^{g}\in R^{s_{1}}$, the undesired output $y^{b}\in R^{s_{2}}$. In a period *t*, the set vectors of input *X* and output *Y* are, respectively, $x_{m}^{t}={\left(x_{1m}^{t},\;x_{2m}^{t},\;\ldots ,\;x_{nm}^{t}\right)}^{T},y_{s}^{t}={\left(y_{1s}^{t},\;y_{2s}^{t},\;\ldots ,\;y_{ns}^{t}\right)}^{T}$. The specific model is as follows:(2)$$\rho^{*}=min\frac{1-\frac{1}{m}\sum_{i=1}^{m}\frac{s_{i}}{x_{i0}}}{1+\frac{1}{s_{1}+s_{2}}\left(\sum_{r=1}^{s_{1}}\frac{s_{r}^{g}}{y_{r0}^{g}}+\sum_{r=1}^{s1}\frac{s_{r}^{g}}{y_{r0}^{g}}+\sum_{i=1}^{s_{2}}\frac{s_{1}^{b}}{y_{i0}^{b}}\right)}$$
(3)$$s.t.\{\begin{array}{c} x_{0}=X\lambda +S^{-}\\ y_{0}^{g}=Y^{g}\lambda -S^{g}\\ y_{0}^{b}=Y^{b}\lambda -S^{b}\\ \overset{n}{\underset{i=1}{\sum }}\lambda_{j}=1,\;\lambda \geq 0;S^{-}\geq 0,\;S^{g}\geq 0,\;S^{b}\geq 0\\ x\in R^{m},\;y^{g}\in R^{s_{1}},\;y^{b}\in R^{s_{2}} \end{array}$$

In the model, $\rho^{*}$ indicates the efficiency value of the research object, $\rho^{*}\in \left[0,\;1\right]$, x indicates the input variable, $S^{-},S_{1}$, $S_{2}$, respectively, indicate the existing input variable, expected output, and undesired output. The slack variables of $y^{g}$, $y^{b}$ denote desired output and undesired output, respectively. Among them, if and only if $\rho^{*}=1$ and $S^{-}=S_{1}=S_{2}=0$, the decision-making unit is efficient, otherwise, the decision-making unit is inefficient.

#### 3.1.3. Coupling Coordination Degree Model

Coupling coordination degree is usually used to indicate the degree of mutual influence and interaction between two or more systems [46]. Based on relevant research, the coupling degree model of agricultural eco-efficiency and food security is established as follows:(4)$$C={\left\{\frac{A\left(x\right)*T\left(y\right)}{\left[A\left(x\right)+T\left(y\right)\right]*\left[A\left(x\right)+T\left(y\right)\right]}\right\}}^{\frac{1}{2}}$$

In the Formula (4), *C* represents the coupling degree between the agricultural eco-efficiency and the food security system, and the value is between 0 and 1.

The coupling coordination degree model helps to further understand the degree of interaction and mutual influence between systems. The basic form of the model is:(5)$$D=\sqrt{C*T},\mathrm{where}\;T=\lambda A\left(x\right)*\mu F\left(y\right)$$

In the Formula (5), *D* is the coupling coordination degree, *C* is the coupling degree, *T* is the comprehensive coordination index between the two systems, and *λ*, *μ* are the undetermined coefficients. In this study, the importance of agricultural eco-efficiency and food security to human livelihood are analyzed. Considering the actual situation and research emphasis, we believe that there is no complete equivalence between agricultural ecological efficiency and food security. Combining their contributions to human development, they are assigned values of 0.3 and 0.7, respectively, and *A*(*x*) and *F*(*y*) are the agricultural eco-efficiency index and food security index, respectively. According to the actual situation and the calculated *D* value, the coupling coordination degree between systems can be divided into five levels [47], as shown in Table 2.

#### 3.1.4. Spatial Autocorrelation Model

(1) Global autocorrelation model. The coupling and coordination degree of agriculture eco-efficiency in various provinces in China may have spatial dependence or autocorrelation at the regional level, which can be measured by the Moran index [48]. The specific formula is:(6)$$I=\frac{\sum_{i=1}^{n}\sum_{j=1}^{n}w_{ij}\left(D_{i}-\bar{D}\right)\left(D_{j}-\bar{D}\right)}{S^{2}\sum_{i=1}^{n}\sum_{j=1}^{n}w_{ij}}$$

In the Formula (6), *I* is the Moran index value, $D_{i}$ is the attribute value of the *i*th spatial unit, and $\bar{D}$ is the mean value of all attribute values of spatial units, $w_{ij}$ is the spatial weight value. $S^{2}$ is the variance, $\sum_{i=1}^{n}\sum_{j=1}^{n}w_{ij}$ is the sum of all spatial weights. The value of the global Moran’s I index is between −1.0 and 1.0. If the value is less than 0, it means that the space is negatively correlated, if it is equal to 0, it means that the space is randomly distributed, and if it is greater than 0, it means that the space is positively correlated.

(2) Local autocorrelation model. In order to reflect the characteristics of the local state, the local space Moran index is used for investigation, and the calculation formula is:(7)$$I_{i}=\frac{\left(D_{i}-\bar{D}\right)}{S^{2}}\sum_{j=1}^{n}w_{ij}\left(D_{j}-\bar{D}\right)$$

$I_{i}$ > 0 indicates positive correlation between zone *i* and surrounding areas, $I_{i}$ < 0 indicates that area *i* is negatively correlated with the surrounding area.

### 3.2. Data

#### 3.2.1. Data Sources

The data scale selected in this paper is from 2011 to 2020, involving 31 provinces in China. Among the basic data involved in the measurement of agricultural eco-efficiency, the data of agricultural labor input, crop sown area, and total output value of agriculture, forestry, animal husbandry and fishery are all from the “China Statistical Yearbook” over the years. The data on agricultural machinery input, effective irrigated area, chemical fertilizers and pesticides and other agricultural inputs are all derived from the “China Rural Statistical Yearbook” and the statistical yearbooks of various provinces. Among the basic data involved in food security are included per capita grain possession, total grain output, sown area of grain crops, proportion of disaster-affected area in total sown area of crops, proportion of expenditure on agriculture, forestry and water affairs, and classification index of grain retail price by region and unit. The data on the amount of water resources used for grain production, the sown area used per unit of grain production, and the basic data required for the calculation of related indicators are all from the “China Statistical Yearbook” and “China Water Conservancy Statistical Yearbook” over the years. Pesticide loss coefficient refers to the “Handbook of Pesticide Loss Coefficient in the First National Survey of Pollution Sources”. The data of regional green food or certified products come from the China Green Food Development Center. Relevant economic indicators have been processed for deflation based on the year 2000. Some missing data were filled by interpolation. Due to differences in data statistics, the study area does not involve Hong Kong, Macau, Taiwan, and other regions.

#### 3.2.2. Measurement of Agricultural Eco-Efficiency and Food Security

The selection of measurement indicators for agricultural eco-efficiency is based on the principle of the existence of input, expected output, and non-expected output, reflecting the relationship between the input level of agricultural inputs and output in agriculture [49,50]. It means that the level of agricultural eco-efficiency is higher, and the selection of specific indicator variables is shown in Table 3. Food security is different from the traditional emphasis on quantity security. In recent years, with the changes in people’s living standards and dietary choices, “food quality and safety” has gradually become one of the important indicators of the new concept of food security [51], and related literature [52,53]; the selection of specific indicator variables is shown in Table 4.

## 4. Results

### 4.1. Analysis of Agricultural Eco-Efficiency

Based on the construction of the carbon emission index system, referring to the carbon emission coefficient of agricultural carbon emission sources [43], according to Formula (1) and Formula (2), through the analysis of MATLAB software, the 31 provinces’ Malmquist index of agricultural eco-efficiency in China from 2011 to 2020 were calculated. Due to space limitations, this paper only lists the results for 2011 and 2020 (Table 5), and analyzes the rate of change.

According to Table 5, it can be seen that in 2020, the average agricultural eco-efficiency in Beijing, Tianjin, Heilongjiang, Fujian, and Hunan was greater than or equal to 1, which belongs to the effective state of agricultural eco-efficiency. The agricultural eco-efficiency in Gansu, Jilin, Shanxi, Inner Mongolia, and other regions is low, and there is still a big gap from efficient emissions. From the perspective of regional differentiation, in 2020, the agricultural eco-efficiency index values in Tianjin, Heilongjiang, Fujian, Guangdong, Guizhou, Yunnan, Tibet, Qinghai, and other regions were significantly higher than those in 2011, indicating that the relevant provinces have made significant progress in reducing agricultural carbon emissions in recent years. It is worth noting that most of these areas belong to provinces with vast terrain or rich ecological resources. With the implementation of the large-scale agricultural land policy, the construction of high standard farmland in the region has been increasing, the input and utilization of agricultural materials have become more efficient, and the protection of ecological resources has achieved excellent results.

However, the agricultural eco-efficiency index has shown a significant downward trend in Shanxi, Inner Mongolia, Liaoning, Jilin, Shanghai, Jiangsu, Zhejiang, Xinjiang, and other provinces, mainly because the development direction of relevant regions is more focused on industry. In the critical stage of transformation, these provinces should pay attention to the improvement of the agricultural scale and standardized production while developing industry and tourism.

### 4.2. Analysis of Food Security Characteristics in China’s Provinces

Based on the construction of the food security index system, the food security indexes of 31 provinces in China from 2011 to 2020 are calculated, as shown in Table 6.

On the whole, the food security level in economically developed areas such as Beijing, Tianjin, Shanghai, Guangdong, and Zhejiang is relatively low, which is mainly related to the regional development strategy formulated by China. The urbanization level of relevant areas is relatively high, focusing on the development of secondary and tertiary industries, the ability to undertake food production is relatively weak. The food security level in Tibet, Guizhou, and Qinghai is relatively low, because the economic development in the relevant areas is relatively backward, the mechanization and scale of agricultural production is low, and the level of cultivated land development is low. Heilongjiang, as one of the important bases of China’s grain production, has fertile soil, a high level of agricultural modernization, and the highest food security index. Anhui, Shandong, Henan and other central regions also undertake the function of ensuring China’s food security and have a relatively high level of food security. It can be seen that the food security level is mainly related to regional planning, and the level of food security in key areas of grain production is relatively high.

### 4.3. Coupling Coordination Degree Analysis

According to Formulas (4) and (5), the coupling coordination degree of agricultural eco-efficiency and food security in each province of China is comprehensively evaluated, and the types of coupling coordination degree are divided at the same time. Due to space limitations, this article only lists the calculation results for 2011 and 2020 (Table 7).

On the whole, most provinces were in a state of low or moderate coordination, and no province was in a state of high coordination in 2011. By 2020, the coupling coordination level in some provinces has been improved, and even a few provinces have reached a state of high coordination. Overall, the degree of coupling coordination between agricultural eco-efficiency and food security in 2020 is generally on the rise compared with 2011, and the comprehensive coordination level has improved to a certain extent, with the average level increasing from 0.413 to 0.440, an increase of 7%. This means that as China attaches great importance to the carbon peaking and carbon neutrality goals and food security, the layout of the agricultural industry and the planning of farmland standards are gradually optimized. While the efficiency of agricultural input and expected output is improved, the effect of food security guarantees has also been improved to a certain extent.

From the perspective of geographical distribution (Table 8), in 2020, Heilongjiang has the highest level of coordination, reaching a high level of coordination, and 11 regions including Hunan, Shandong, Henan, Anhui, and Jiangsu have reached moderate coordination, while 12 regions including Beijing, Tianjin, Shanxi, Shanghai, and Zhejiang are still in a state of low coordination. In general, the provinces with a higher coupling coordination degree are mostly distributed in areas with relatively developed high standard farmland construction and the central region with large grain production scale, while the coupling coordination degree of Beijing, Shanghai, Guangzhou, and coastal economically developed areas is limited by agricultural resources. The restriction of endowment is different from the division of labor in the overall national planning, insufficient large-scale production, and insufficient attention to food production, resulting in a low level of overall coordination between agricultural eco-efficiency and food security.

From the perspective of coupling coordination types, it can be divided into five categories: low coordination efficiency lag type, low coordination food security lag type, moderate coordination efficiency lag type, moderate coordination food security lag type, and high coordination food security lag type. The results are mainly moderately coordinated. In 2020, the number of such provinces increased from 16 in 2011 to 19. This shows that, although the overall coupling and coordination level of each province needs to be improved, in recent years, with the shift of policy calls and development focus, the agricultural eco-efficiency and food security system are gradually developing towards a high level. The type of moderately coordinated food security lagging behind represents the majority among the moderately coordinated provinces, accounting for an average of 41.94% in 2020, but the number has decreased compared with 2011, a year-on-year decrease of 9.67%.The second is the low coordination food security lagging type, which will account for an average of 35.48% in 2020, a year-on-year decrease of 9.68% from 2011. It shows that most provinces have achieved further stability in protecting cultivated land area, building high standard farmland, and ensuring grain production, and the comprehensive grain production capacity under the guidance of food security and cultivated land protection policies.

Focusing on each province, Gansu has changed from a low-level coordinated food security lagging type in 2011 to a low-level coordination efficiency lagging type in 2020, and food security efficiency and green and clean production still needs to be strengthened. Eleven regions, including Beijing, Tianjin, Shanxi, Shanghai, and Zhejiang, belong to the low-level coordinated food security lagging type, which may be related to the national grain production development planning and division of labor. Some developed areas and eastern coastal areas undertake important economic growth functions. The level of industrialization is relatively high, and insufficient attention has been paid to food security.

Five regions including Jilin, Inner Mongolia, Anhui, Henan, and Shandong belong to the moderate coordination efficiency lagging type, and the relevant regions belong to the main grain-producing areas of the country, and mainly undertake the function of ensuring food security. However, the mismatch between grain production and economic development has led to the agricultural planting in many major grain-producing regions still staying in traditional grain crops. New agricultural technologies and equipment have yet to penetrate into various fields of agricultural production, and production capacity urgently needs to leapfrog to a new level. Thirteen regions, including Liaoning, Xinjiang, Shaanxi, Hebei, and Jiangxi, belong to the moderately coordinated food security lagging type, which may be due to the close relationship between food planting and natural conditions. In some areas, there are desertification and salinization phenomena, and the cultivated land resources are limited, the facilities are relatively backward, and the planting layout needs to be optimized. Heilongjiang belongs to the highly coordinated food security lagging type. The high agriculture eco-efficiency is mainly due to the province’s high level of land scale and intensification, but the grain production capacity still needs to be further improved. To sum up, the main grain-producing areas in central China bear the effect of guaranteeing food security, but the traditional extensive agricultural production mode needs to be improved urgently. China should continue to strengthen the development of agricultural clean production to achieve the goal of reducing agricultural carbon emissions. The effect of food security in some economically backward areas needs to be improved. Efforts should be made to improve the quality of cultivated land, increase agricultural productivity, and strengthen the ability to ensure food security.

### 4.4. Spatial Effect Analysis

#### 4.4.1. Global Spatial Autocorrelation Analysis

The paper uses Stata software to calculate the global Moran index of the coupling coordination degree of agricultural eco-efficiency and food security in each province from 2011 to 2020, as shown in Table 9.

The measurement results of global spatial correlation under geographical matrix (W_g_) show that the Moran index of each province in China has a significant spatial positive correlation in all years except 2015 under the W_g_, and the significance level has increased from 2017 to 2020, but the overall correlation level is not high, which means that the spatial agglomeration effect of coupling coordination has increased in recent years, but the overall level of interaction between provinces is weak. The reason may be due to the differences in the factors of agricultural resource endowment and the degree of modern agricultural development in different regions: On the one hand, according to the characteristics of regional natural resources and endowments, some provinces have a higher level of high standard basic farmland construction, a higher degree of scale and mechanization, and a better development of agricultural eco-efficiency, leading to a large gap in development levels among provinces. On the other hand, China’s provinces have positive spatial dependence on adjacent regions, but the effect is relatively weak, which leads to a large gap in the growth rate of the coupling coordination level.

#### 4.4.2. Local Spatial Autocorrelation Analysis

In order to further illustrate the spatial correlation characteristics of the coupling coordination degree of 31 provinces in China from 2011 to 2020, 2011, 2016, and 2020 were selected as the research years to draw the local Moran index map of their coupling and coordination development status (due to space constraints and scatter map, it is difficult to identify each province, so the scatter map is not presented here); the results are shown in Table 10.

It can be seen from the table that the high–high agglomeration areas in 2011, 2016, and 2020 included seven provinces: Heilongjiang, Shandong, Jilin, Liaoning, Henan, Anhui, and Hubei. The agricultural resource endowment of these provinces is generally good, especially in the northeastern region, where the soil is fertile, the terrain is flat, the level of agricultural modernization is relatively high, and the synergy between agricultural eco-efficiency and food security is strong. In 2020, Chongqing and Hunan joined the high–high agglomeration area, which may be driven by the implementation of policies in recent years and the spillover effects of surrounding areas. In 2011, the low–low agglomeration area included 13 provinces including Beijing, Tianjin, Qinghai, Hainan, Ningxia, Gansu, Tibet, Fujian, Yunnan, Zhejiang, Guangdong, Guangxi, and Shaanxi.

From the perspective of regional development orientation, Beijing, Tianjin, Hainan, Zhejiang, Guangdong, and other regions do not mainly undertake the function of food security, and the level of food security is relatively low, but the economy is relatively developed, the level of science and technology is high, and the agricultural eco-efficiency is high. The level of coordination between agricultural eco-efficiency and food security is low. The reason for Shanghai’s transition from a low–high agglomeration area to a low–low agglomeration area is also the same in 2020. In the same year, Xinjiang changed from a high–low agglomeration area to a low–low agglomeration area. The reason may be that with the government’s emphasis on the overall ecological environment, the agricultural eco-efficiency has been impacted in the short term, and may improve in the long term.

## 5. Conclusions

Based on the panel data of 31 provinces in China, firstly, the undesirable output SBM model is used to comprehensively measure the agricultural eco-efficiency and food security level of each province in China, and then the coupling coordination degree model is used to measure the coupling coordination level of agricultural eco-efficiency and food security. Finally, the spatial correlation between agricultural eco-efficiency and food security was validated using a spatial econometric model. The results show that: first, on the whole, most provinces were in a state of low coordination or moderate coordination in 2011, and no province showed a high coordination state. By 2020, the level of coupling coordination in some provinces has improved, and a few provinces have even reached a highly coordinated state. Second, economically developed cities such as Beijing, Shanghai, and Tianjin, and remote areas such as Tibet, Qinghai, and Ningxia are mostly of the lagging food security type, and Jilin, Shandong, Henan and other major grain production areas are mostly of the lagging agricultural eco-efficiency type. Third, the coupling coordination level between agricultural eco-efficiency and food security shows a positive spatial correlation in terms of geographical distance, and the positive spatial dependence gradually increases but is not strong overall.

The research in this paper is of great significance for promoting the dual goals of agricultural eco-efficiency and food security. Based on the main conclusions, the following policy recommendations are put forward:

(1) Increase agricultural technology R&D and promotion, and provide high utilization rates of agricultural resources such as chemical fertilizers and pesticides. First, we should strengthen cooperation with scientific research institutes, coordinate the targeted training of required professional and technical personnel, and provide reserve forces for agricultural technology research and development and promotion. The second is to strengthen the innovation of agricultural inputs, such as innovating the fermentation technology of organic fertilizers, and strengthening scientific research on acidified and salinized land caused by excessive fertilization. Multiple measures should be taken for increasing the utilization rate of investment in agricultural materials such as grain crops, fertilizers and pesticides, reducing the residues of fertilizers and pesticides, and promoting the development of ecological agriculture;

(2) Promote the construction of high standard farmland and innovate various forms of production models. Agricultural production models should be actively explored that can not only solve environmental problems, but also ensure food security. The importance should be fully understood of high standard farmland construction to increase the yield of cultivated land indicators, enrichment of production methods according to local conditions, building a number of modern agricultural industrial parks for grain, extending the industrial chain, focusing on building distinctive brands, continuously improving the comprehensive benefits of farmland, and improving the efficiency of farmland utilization and agricultural productivity. It is also important to avoid the waste of rural resources, form a scientific and environmentally friendly cycle chain, and achieve multi-efficiency win–win under the ecological background;

(3) Strengthen the inter-regional coordinated development mechanism and enhance the spatial spillover effect. The main grain-producing areas such as the Northeast and Central China are the backbone of grain production, while Beijing and the eastern coastal areas are more developed in economy, have a higher level of science and technology, and have higher agricultural eco-efficiency. We should actively break the barriers of inter-regional development, give full play to the role of technological leadership and demonstration in economically developed regions, raise the awareness of agricultural green production in major grain-producing areas, comprehensively promote modern agricultural production technologies, and promote green and low-carbon transformation of agriculture.

## Figures and Tables

**Figure 1 ijerph-20-00431-f001:**
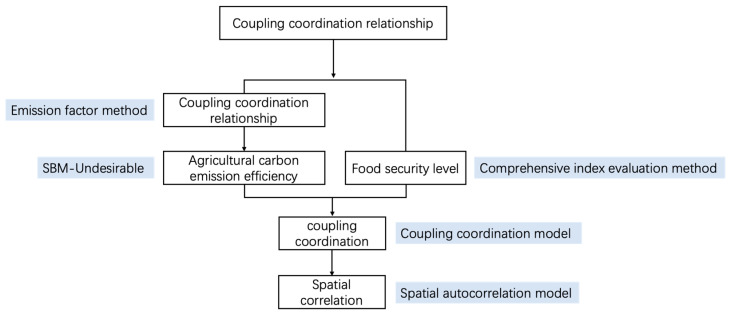
Methodology steps diagram.

**Table 1 ijerph-20-00431-t001:** Agricultural carbon emission sources, coefficients and reference sources.

Carbon Source	Carbon Emission Factor	Reference Source
fertilizer	0.896 kgC/kg	Oak Ridge National Laboratory [42]
pesticide	4.934 kgC/kg	Oak Ridge National Laboratory
agricultural film	5.180 kgC/kg	Institute of Agricultural Resources and Ecological Environment, Nanjing Agricultural University
diesel fuel	0.593 kgC/kg	2013 IPCC United Nations Intergovernmental Committee of Experts on Climate Change [43]
irrigation	266.48 kgC/hm^2^	Reference related literature [44]
ploughing	312.6 kgC/km^2^	Reference related literature

**Table 2 ijerph-20-00431-t002:** Coupling coordination degree grading system.

*D*	Grading	Index
(0.0, 0.2]	Severe maladjustment	
(0.2, 0.4]	Low coordination	X<Y, Efficiency lagging type
(0.4, 0.6]	Moderate coordination	X=Y, Efficiency lagging type
(0.6, 0.8]	High coordination	X>Y, Lagging food security
(0.8, 1.0]	Extreme coordination	

**Table 3 ijerph-20-00431-t003:** The Undesirable Output SBM Model Index System for Measuring Agricultural Eco-Efficiency.

Index	Category	Variable	Units	Explanation
Input	Labor input	Labor	10^4^	Proportion of rural individual employment
	Capital input	Machinery	10^4^ kw	Agricultural machinery investment
		Irrigation	10^4^ hm^2^	Effective irrigation area
	Land input	Sown area	10^4^ hm^2^	Crop sown area
	Agricultural inputs	Chemical fertilizer	10^4^ kg	Fertilizer application amount
		Pesticide	10^4^ kg	Pesticide application amount
		Agricultural film	10^4^ kg	Amount of agricultural plastic film
		Diesel	10^4^ kg	Application amount of agricultural diesel
Output	Desirable output	Total output value of agriculture, forestry, animal husbandry and fishery	10^8^ yuan	Total grain output
	Undesirable output	Agricultural carbon emissions	10^4^ kgc	Total agricultural carbon emissions

**Table 4 ijerph-20-00431-t004:** Index system for measuring food security.

Target	Criterion	Explanation	Units	Attribute
Food security	Quantity security	Per capita share of grain	tons/per	+
		Total grain output	10^4^ tons	+
		Sown area of grain crops	10^4^ hm^2^	+
		Proportion of disaster affected area in total planting area of crops	%	−
	Quality safety	Pesticide loss coefficient * pesticide usage/grain crop yield	tons/tons	−
		Number of certified green food products by region in the year	Unit	+
	Economic security	Agriculture, forestry and water affairs expenditure/local public finance expenditure	%	+
		Sub index of grain retail price by region	−	−
	Resource security	Water resources used per unit grain output	m^3^/ton	−
		Sown area per unit grain yield	hm^2^/ton	−

**Table 5 ijerph-20-00431-t005:** Agricultural eco-efficiency and ranking of provinces in China.

Area	2011		2020		Rate%
	Index	Rank	Index	Rank	
Beijing	1.186	2	1.076	9	−9.22
Tianjin	1.072	13	1.503	1	40.30
Hebei	0.573	25	0.413	25	−27.93
Shanxi	0.423	30	0.345	29	−18.25
Inner Mongolia	1.107	10	0.364	28	−67.11
Liaoning	1.068	14	0.537	20	−49.72
Jilin	0.628	21	0.289	30	−54.04
Heilongjiang	0.517	28	1.037	11	100.54
Shanghai	1.074	12	0.420	24	−60.93
Jiangsu	1.010	17	0.596	17	−41.00
Zhejiang	1.024	15	0.520	21	−49.24
Anhui	0.529	27	0.405	27	−23.43
Fujian	1.096	11	1.117	8	1.88
Jiangxi	0.578	24	0.557	18	−3.62
Shandong	0.608	22	0.412	26	−32.24
Henan	0.594	23	0.472	22	−20.44
Hubei	0.732	20	0.625	16	−14.58
Hunan	1.153	5	1.031	12	−10.61
Guangdong	1.139	8	1.152	7	1.16
Guangxi	1.123	9	1.051	10	−6.43
Hainan	1.401	1	1.289	3	−7.97
Chongqing	1.021	16	1.001	14	−1.93
Sichuan	1.140	7	0.860	15	−24.56
Guizhou	1.009	18	1.325	2	31.34
Yunnan	0.500	29	1.001	13	100.23
Tibet	1.001	19	1.200	4	19.88
Shaanxi	1.164	3	1.161	6	−0.25
Gansu	0.330	31	0.281	31	−14.86
Qinghai	1.146	6	1.199	5	4.60
Ningxia	0.544	26	0.436	23	−19.88
Xinjiang	1.157	4	0.551	19	−52.32

**Table 6 ijerph-20-00431-t006:** Food security and ranking of provinces in China.

Area	2011		2020		Rate%
	Index	Rank	Index	Rank	
Beijing	0.081	30	0.081	31	−0.14
Tianjin	0.089	29	0.103	30	15.58
Hebei	0.402	4	0.391	10	−2.61
Shanxi	0.184	21	0.283	18	53.94
Inner Mongolia	0.328	9	0.523	5	59.53
Liaoning	0.259	12	0.288	16	11.16
Jilin	0.360	7	0.483	7	34.24
Heilongjiang	0.598	1	0.925	1	54.73
Shanghai	0.077	31	0.161	25	108.86
Jiangsu	0.383	5	0.497	6	29.74
Zhejiang	0.187	19	0.210	22	12.32
Anhui	0.371	6	0.600	3	61.76
Fujian	0.154	25	0.159	26	3.33
Jiangxi	0.259	13	0.324	14	25.02
Shandong	0.512	2	0.661	2	29.07
Henan	0.461	3	0.599	4	29.95
Hubei	0.322	10	0.389	11	20.91
Hunan	0.318	11	0.469	8	47.41
Guangdong	0.183	22	0.175	24	−4.33
Guangxi	0.200	17	0.251	19	25.52
Hainan	0.089	28	0.110	29	23.93
Chongqing	0.187	20	0.359	12	91.91
Sichuan	0.354	8	0.419	9	18.49
Guizhou	0.158	23	0.227	20	43.42
Yunnan	0.252	14	0.339	13	34.61
Tibet	0.128	26	0.146	27	13.92
Shaanxi	0.192	18	0.223	21	15.94
Gansu	0.204	15	0.313	15	53.23
Qinghai	0.091	27	0.139	28	53.24
Ningxia	0.158	24	0.187	23	18.57
Xinjiang	0.203	16	0.287	17	41.57

**Table 7 ijerph-20-00431-t007:** The coupling coordination degree and comprehensive evaluation of agricultural eco-efficiency and food security in various provinces of China.

Area	2011			2020			Rate%
	Index	Rank	Type	Index	Rank	Type	
Beijing	0.318	30	②	0.311	31	②	−2.11
Tianjin	0.319	29	②	0.358	27	②	12.13
Hebei	0.472	8	④	0.446	13	④	−5.58
Shanxi	0.343	24	②	0.388	22	②	13.11
Inner Mongolia	0.486	6	④	0.484	9	③	−0.44
Liaoning	0.446	12	④	0.416	19	④	−6.76
Jilin	0.460	9	④	0.453	12	③	−1.50
Heilongjiang	0.535	1	③	0.692	1	⑤	29.31
Shanghai	0.307	31	②	0.327	30	②	6.39
Jiangsu	0.505	3	④	0.512	6	④	1.44
Zhejiang	0.398	18	②	0.370	26	②	−7.00
Anhui	0.454	10	④	0.515	5	③	13.56
Fujian	0.379	21	②	0.384	23	②	1.30
Jiangxi	0.405	16	④	0.436	16	④	7.66
Shandong	0.519	2	④	0.534	3	③	2.84
Henan	0.499	5	④	0.528	4	③	5.87
Hubei	0.453	11	④	0.473	11	④	4.48
Hunan	0.484	7	④	0.544	2	④	12.37
Guangdong	0.403	17	④	0.398	20	②	−1.17
Guangxi	0.413	14	④	0.440	15	④	6.56
Hainan	0.338	27	②	0.354	28	②	4.67
Chongqing	0.398	19	②	0.493	8	④	23.92
Sichuan	0.501	4	④	0.509	7	④	1.57
Guizhou	0.376	22	②	0.443	14	④	17.86
Yunnan	0.393	20	②	0.484	10	④	23.17
Tibet	0.352	23	②	0.379	24	②	7.69
Shaanxi	0.410	15	④	0.430	17	④	4.86
Gansu	0.343	25	②	0.389	21	①	13.54
Qinghai	0.326	28	②	0.374	25	②	14.71
Ningxia	0.338	26	②	0.347	29	②	2.49
Xinjiang	0.418	13	④	0.417	18	④	−0.10
Average	0.413	—	—	0.440	—	—	7.00

According to the actual measurement results of different provinces, they can be divided into five types: ① Low coordination efficiency hysteresis; ② Low Coordination Food Security Lag Type; ③ Moderate Coordination Efficiency Lag Type; ④ Moderate coordinated food security lagging type; ⑤ Highly coordinated food security lagging type.

**Table 8 ijerph-20-00431-t008:** Spatial distribution of the coupled and coordinated development of agricultural eco-efficiency and food security in 31 provinces from 2011 to 2020.

Coupling Coordination Level	2011	2020
Severe disorder	—	—
Low coordination	Beijing, Tianjin, Shanxi, Shanghai, Zhejiang, Fujian, Hainan, Chongqing, Guizhou, Yunnan, Tibet, Gansu, Qinghai, Ningxia	Beijing, Tianjin, Shanxi, Shanghai, Zhejiang, Fujian, Guangdong, Hainan, Tibet, Qinghai, Gansu, Ningxia
Moderate coordination	Hebei, Inner Mongolia, Liaoning, Jilin, Jiangsu, Anhui, Jiangxi, Shandong, Henan, Hubei, Hunan, Guangdong, Guangxi, Sichuan, Shaanxi, Xinjiang, Heilongjiang	Hebei, Inner Mongolia, Liaoning, Jilin, Jiangsu, Anhui, Jiangxi, Shandong, Henan, Hubei, Hunan, Guangxi, Chongqing, Sichuan, Guizhou, Yunnan, Shaanxi, Xinjiang
High coordination	—	Heilongjiang
Extreme coordination	—	—

**Table 9 ijerph-20-00431-t009:** Global spatial correlation.

Year	W_g_	Year	W_g_
2011	0.143 * (1.459)	2016	0.135 * (1.443)
2012	0.153 * (1.548)	2017	0.164 ** (1.710)
2013	0.155 * (1.564)	2018	0.158 ** (1.705)
2014	0.154 * (1.567)	2019	0.153 * (1.606)
2015	0.111 (1.227)	2020	0.210 ** (2.113)

Note: **, and * indicate significance at the 5%, and 10% levels, respectively.

**Table 10 ijerph-20-00431-t010:** Local spatial clustering of coupling coordination levels in 31 provinces in main years.

Year	High–High	Low–High	Low–Low	High–Low
2011	Heilongjiang, Shandong, Jilin, Liaoning, Henan, Anhui, Hubei	Shanghai, Shanxi, Guizhou, Chongqing, Jiangxi	Beijing, Tianjin, Qinghai, Hainan, Ningxia, Gansu, Tibet, Fujian, Yunnan, Zhejiang, Guangdong, Guangxi, Shaanxi	Sichuan, Xinjiang, Hunan, Jiangsu, Inner Mongolia, Hebei
2016	Heilongjiang, Shandong, Jilin, Liaoning, Henan, Anhui, Hubei	Shanghai, Shanxi, Guizhou, Chongqing, Jiangxi, Tibet, Yunnan, Guangxi	Beijing, Tianjin, Qinghai, Hainan, Ningxia, Gansu, Fujian, Zhejiang, Guangdong, Shaanxi	Sichuan, Xinjiang, Hunan, Jiangsu, Inner Mongolia, Hebei
2020	Heilongjiang, Shandong, Jilin, Henan, Anhui, Hubei, Chongqing, Hunan	Shanxi, Guizhou, Jiangxi, Tibet, Liaoning, Shaanxi, Guangxi	Beijing, Tianjin, Qinghai, Hainan, Ningxia, Gansu, Fujian, Zhejiang, Guangdong, Shanghai, Xinjiang	Sichuan, Yunnan, Inner Mongolia, Jiangsu, Hebei

## Data Availability

Data openly available in a public repository. All the data that support the findings of this study are openly available at [https://data.cnki.net].

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
