# Peer review of "Coupling and Coordinating Relationship between Agricultural Eco-Efficiency and Food Security System in China"

_ijerph, 2022, doi:10.3390/ijerph20010431_

Round 1

Reviewer 1 Report

It is an interesting topic to study relationship between agricultural eco-efficiency and food security systems for China's agricultural carbon mitigation policy.  Some details need further correction. (1) Add the complete calculation process for agricultural carbon emissions in your study, because it is a basis for this paper, including carbon sources and related coefficients. (2) In table 2, why you choose the variable-pesticide in the  pesticide application rate instead of amount? besides, why you choose total output value of agriculture, forestry,animal husbandry and fishery as the desirable output, instead of total grain output?  Otherwise, it will lead to the inaccuracy. Please recorrect it in your revised version. (3) It will be more complete from the perspective of sutructure based on the above conclusions if you further  take a disscussion (4) English language and style are minor spell check required.

Reviewer 2 Report

This paper explored the synergistic relationship between agricultural eco-efficiency and food security systems based on some methods, which has practical significance for China's agricultural development. However, there are some shortcomings that need major revision. I put forward some suggestions as follows.

1. The first sentence of the abstract does not seem to highlight the importance of the paper's research, and the statement of “displaying the discord of co-corruption” is puzzled.

2. The background of the study in the introduction emphasizes the relationship between agricultural food security and Greenhouse gas emissions, which seems to have little relationship with the topic of this paper, and it is not clear why the authors want to study the coordinated coupling of agricultural eco-efficiency and food security systems.

3. The explanation of the letters in Equation 1 and Equation 2 is incomplete and needs to be completed.

4. Authors should at least indicate the spatial scale of the study and the time period of the study in 3.2 Data; it is also very important that we know the carbon emission data is not directly obtained, please give the detailed calculation process of agricultural carbon emission.

5. Line 181 should preferably be marked with the cited reference.

6. In calculating the coupling coordination degree, the authors assign calculated values of 0.3 and 0.7 to agricultural eco-efficiency and food security respectively, on what basis? Why not 0.5 and 0.5?

7. In Table 1, both X=Y and X>Y are efficiency lagging type, please do further refinement according to the notes in Table 5 and give the basis for dividing these five categories; the unit of labor output is further specified.

8. In 3.4 Spatial autocorrelation, the formula and explanation of local autocorrelation model are missing; the letters appearing in equation 5 do not correspond to the letters in line 200, and there are obvious errors in the formula; specify the type of spatial weight matrix in spatial autocorrelation, please.

9. In terms of writing order, authors are advised to place the methods and empirical results of agricultural eco-efficiency and food security in front of the coupling coordination degree.

10. The title of 3.5 is not appropriate, and it is suggested that it be changed to measurement of agricultural eco-efficiency and food security.

11. There are three ways to write about agricultural eco-efficiency in the paper, such as: Agricultural Eco-Efficiency, Agricultural Ecological Efficiency, and agroecological efficiency, which are suggested to be unified

12. There is an obvious error in the first sentence of 4.1, the paper does not construct a carbon emission index system; the second paragraph of the analysis on page 7 is on agricultural eco-efficiency instead of agricultural carbon emission, please check further.

13. Table 4 does not reflect the process of decomposition, and the title is wrong.

14. Some of the explanations in the paper do not explain the reasons well and lack comparative discussion with the findings of the relevant literature, for example:

Lines 246-251 "may be related to the development orientation" needs to be further expanded.

Lines 283-285 "Heilongjiang has the highest level of coordination" needs to be further analyzed.

15. The article should add a table for food security in between Tables 4 and 5 to better understand the specific divisions of Table 5.

16. Note should be given under Table 7, what **, * and in brackets stand for respectively.

17. In the global Moran analysis in 4.3.1, it is suggested that the authors better delete the part of the economic weight matrix, which is not significant and has no economic significance; the local Moran in 4.3.2 does not state clearly which spatial weight matrix is used, and I do not understand why the authors cannot represent it graphically, and why it is difficult to identify each province.

18. In this paper, the authors confuse spatial autocorrelation, which is mainly used to reveal spatial dependence rather than spatial spillover, and spatial econometric models, which can generally explain spatial spillover effects. In the first paragraph of the conclusion section, spatial autocorrelation is described as a spatial econometric model.

19. Policy implications need to be improved to avoid vague and contradictory recommendations. For example, statements such as “planting + breeding + processing", "food crops + economic crops + ecological crops", "planting + biogas" does not match the conclusion; “the intermediate economic value created every year is as high as 100 billion yuan. (...), which is still far from the 50% to 60% of developed countries in Europe and the United States” are not recommendations.

20. Please double check the English grammar.

Some minor issues:

1.155 lines are mispunctuated, 46 lines have no punctuation at the end of the sentence, and 226 lines are semantically incoherent;

2. inconsistent formatting of references, such as names in 14, extra symbols at the beginning of 45;

3. The first letter of Research in the 8th line of the abstract should be lowercase.

Reviewer 3 Report

This manuscript found that the higher level of coupling coordination always occurs in those areas with high-standard farmland construction and large grain production scale, while economically developed areas appear to be a lower overall coordination level limited by endowment constraints and division of labor in development planning. The manuscript has been written well.  However, the following concerns needed to revise.

In the introduction, authors are recommended to add a list of contributions before the outlined paragraph.

In related work, please provide a comparative table to show the limitation and advantages of relevant studies.

Add numbering results in the abstract section

Section of Data and Methodology, please separate into two sections.

Provide a flowchart for your methodology steps

Why is this study focusing on Agricultural  Ecological Efficiency and Food Security Systems?

Round 2

Reviewer 1 Report

As you have answered in your response,  "due to the availability of data, the selected relevant input indicators, such as fertilizer and pesticide, are all based on the cultivated land area, rather than the grain planting area. Therefore, in order to improve the validity and scientificity of the data, the gross output value of agriculture, forestry, animal husbandry and fishery is selected for calculation." In fact, almost all relevant indicators for quantity security or resources security in Table 3 and 4, only related to  grain production, which is inconsistent with your indicator for economic security. How to explain it?  Why you don't  select more suited economic indicator for more exact analysis ?

Author Response

Once again, I would like to express my sincere gratitude for your suggestions.

Table 3 is the measurement of agricultural eco-efficiency, and indicators are selected from the overall agricultural level. Table 4. Index system for measuring food security. This table comprehensively measures the level of food security from the dimensions of quantity security, quality security, economic security and resource security.

Based on the comprehensive calculation of the above two systems, explore the correlation and synergy between agricultural ecological efficiency and food security.

Thanks for the suggestions, and I hope that in the future, we can conduct further field investigations to obtain data at the micro level and make the research more scientific.

Reviewer 2 Report

I believe the author has answered my questions well, so I suggest publishing this manuscript with minor revisions.

1. In line 208, the authors assign values of 0.3 and 0.7 to agricultural eco-Efficiency and food security, still without giving the basis for the assignment.

2. References 43 and 54 are duplicated.

3. Reference 14 is by two authors, please check it again. There is also a problem with the author format of reference 43.

Author Response

Dear reviewers and editors:

Re: Manuscript ID: ijerph-2100753 and Title: Coupling and Coordinating Relationship between Agricultural Ecological Efficiency and Food Security System in China

We should love to thank you for allowing us to resubmit a revised copy of the manuscript and we highly appreciate your time and consideration. The responses to the reviewer’s comments are marked in red and presented following.

Sincerely.

Ruixue Wang.

Q1. In line 208, the authors assign values of 0.3 and 0.7 to agricultural eco-Efficiency and food security, still without giving the basis for the assignment.

Response: Thank you for your suggestion. We further explained this part.

Q2. References 43 and 54 are duplicated.

Response: We deeply appreciate the suggestion, and we deleted duplicate literature.

Q3. Reference 14 is by two authors, please check it again. There is also a problem with the author format of reference 43.

Response: We agree with your suggestion and have revised and supplemented References 14 and 43.

Reviewer 3 Report

Done.

Author Response

Thank you again! I wish you success in your work and good health.